# Quantifying evolutionary importance of protein sites: A Tale of two measures

**Avital Sharir-Ivry**, **Yu Xia***

Department of Bioengineering, McGill University, Montreal, Quebec, Canada

* brandon.xia@mcgill.ca

## Abstract

A key challenge in evolutionary biology is the accurate quantification of selective pressure on proteins and other biological macromolecules at single-site resolution. The evolutionary importance of a protein site under purifying selection is typically measured by the degree of conservation of the protein site itself. A possible alternative measure is the strength of the site-induced conservation gradient in the rest of the protein structure. However, the quantitative relationship between these two measures remains unknown. Here, we show that despite major differences, there is a strong linear relationship between the two measures such that more conserved protein sites also induce stronger conservation gradient in the rest of the protein. This linear relationship is universal as it holds for different types of proteins and functional sites in proteins. Our results show that the strong selective pressure acting on the functional site in general percolates through the rest of the protein via residue-residue contacts. Surprisingly however, catalytic sites in enzymes are the principal exception to this rule. Catalytic sites induce significantly stronger conservation gradients in the rest of the protein than expected from the degree of conservation of the site alone. The unique requirement for the active site to selectively stabilize the transition state of the catalyzed chemical reaction imposes additional selective constraints on the rest of the enzyme.

**Data Availability Statement:** Analysis scripts are available from https://github.com/AvitalSharirIvry/Quantifying-Evolutionary-Importance-of-Protein-Sites-A-Tale-of-Two-Measures.git.

**Funding:** This work was supported by Natural Sciences and Engineering Research Council of

## Author summary

Sites within proteins which are important for stability or function are under stronger selective pressure and evolve more slowly than other sites. Catalytic sites in enzymes are such highly conserved sites with relatively low evolutionary rates. Recently, catalytic sites were shown to induce a strong gradient of conservation such that the closer a residue is to the catalytic site, the more conserved it is. Here we show that there is a universal linear relationship between the degree of evolutionary conservation of a protein site and the conservation gradient it induces in the protein tertiary structure, applicable to all types of sites. Our findings suggest that selective pressure acting on a protein site generally percolates through the rest of the protein via residue-residue contacts. Remarkably however, catalytic sites induce significantly stronger conservation gradients than expected from their degree of conservation alone. Our results indicate that the strong conservation gradient induced by catalytic sites is driven by the unique function of enzyme catalysis, which

Canada grant numbers RGPIN-2019-05952, RGPAS-2019-00012 (Y.X.) (https://www.nserc-crsng.gc.ca/index_eng.asp), Canada Foundation for Innovation grant number JELF-33732 (Y. X.) (https://www.innovation.ca/), and Canada Research Chairs program (Y. X.) (https://www.chairs-chaires.gc.ca/home-accueil-eng.aspx). The funders had no role in study design, data collection and analysis, decision to publish, or preparation of the manuscript.

**Competing interests:** The authors have declared that no competing interests exist.

requires the participation of many residues beyond the few key catalytic residues. Our results provide insights into evolutionary conservation patterns of and surrounding proteins functional sites, with implications for functional site prediction and protein design.

## Introduction

The evolutionary importance of protein sites under purifying selection can be quantified in two very different ways. The classical, "intrinsic" measure for the evolutionary importance of a protein site is the degree of conservation or evolutionary rate of the protein site itself. Protein residues experience different degrees of selective pressure as a result of the different roles they play in protein stability and function[1,2]. For example, residues in a protein core are generally under stronger selective pressure than surface residues due to their importance in stabilizing the protein. Indeed, structural determinants such as solvent exposure[3–7] and degree of packing [8–10] were shown to explain a large portion of the variability in the observed site-specific evolutionary rates. In addition, residues in functional sites such as catalytic sites[11] and ligand-binding sites are also under stronger selective pressure than non-functional residues.

An alternative, "extrinsic" measure for the evolutionary importance of a protein site is the conservation gradient the site exerts on the rest of the protein. Rather than quantifying evolutionary conservation of the protein site itself, this measure captures how the evolutionary conservation of other residues surrounding the site gradually decreases with distance from the site in the tertiary structure. Several studies have indicated the possibility for selective pressure to propagate from the functional site to the rest of the protein via physical interactions between neighboring residues in the three-dimensional structure [12–15]. While it is clear that the two measures of the evolutionary importance of protein sites are substantially different, it remains unknown how the two measures relate to each other.

In this paper, we addressed the fundamental question whether there is a direct relationship between the intrinsic and extrinsic measures of the evolutionary importance of protein sites. We have based our study on a dataset of homology-based structural models of the yeast proteome [5,7]. Despite their major differences, we show here that there is a strong linear relationship between the degree of conservation of a protein site and the conservation gradient induced from it. In other words, more conserved protein sites tend to induce stronger conservation gradient in the rest of the protein, as selective pressure acting on the protein site percolates via residue-residue interactions. This linear evolutionary conservation-percolation relationship is universal in that it holds for different types of proteins as well as for different types of functional sites in proteins. Remarkably however, catalytic sites in enzymes are the exception to this universal rule, as catalytic sites induce significantly stronger conservation gradient than other types of functional sites with similar degrees of conservation. We conclude that for many different types of functional sites, site-induced conservation gradient can be explained by the percolation of site-specific selective pressure through the rest of the protein via residue-residue contacts. However, catalytic sites in enzymes induce significantly stronger conservation gradient in the rest of the protein than expected from the percolation theory. This is likely due to the unique requirement for the enzyme active site to selectively bind to and stabilize the transition state of the catalyzed chemical reaction [16].

Overall, we show that a more complete understanding of the selective pressure on protein sites can be achieved by integrating the intrinsic measure of site-specific evolutionary conservation with the extrinsic measure of site-induced conservation gradient, with potential implications in protein design, functional site prediction and the study of disease mutations.

## Results

### Evolutionary conservation gradient induced from a protein residue in the proteins structure is linearly correlated with the conservation of the residue

We based our study on a dataset of homology-based structural models of the *S. cerevisiae* proteome. This basic dataset contains structural templates from the Protein Data Bank (PDB)[17] mapped to ORFs of *S. cerevisiae* via sequence alignment (see Methods). We had 1274 yeast ORFs with structural models from the PDB for which residue conservation scores are available in ConSurf-DB [18,19]. Each residue in the dataset was ranked according to its relative conservation within the protein (the residue's rank of conservation divided by the total number of residues). This normalized conservation rank of a residue ranges from 0 to 1, with higher rank corresponding to higher conservation within the protein. We have also calculated for each residue the Pearson correlation between the conservation scores of all other residues in the protein and their distance from that reference residue. This Pearson correlation between conservation and distance describes the degree of percolation of the evolutionary conservation from a residue throughout the protein tertiary structure, i.e., the 'conservation gradient'.

A clear negative linear trend is observed between the conservation rank of a residue and the strength of the evolutionary conservation gradient induced from it (Fig 1A). The more a residue is conserved within the protein (higher conservation rank), the stronger the evolutionary conservation gradient it induces. Similar result is shown when conservation gradients are calculated using Spearman correlation rather than Pearson correlation (S1A Fig). Conservation gradients calculated up to 30Å away from the residue, are overall lower compared to those over the entire domain, however they exhibit higher correlation to residue conservation (S2A and S3A Figs). The distribution of overall per-protein Pearson correlations between conservation ranks and conservation gradients (per-protein conservation-percolation trend) is mainly between -0.2 and -0.7, and the average correlation in the dataset is -0.5 (Fig 1B), indicating that the linear relationship between site-specific conservation and site-induced conservation gradient is high for all types of proteins. Similarly, with conservation gradients calculated with Spearman correlation, the average per-protein correlation between conservation gradient and conservation rank is -0.5 as well (S1B Fig). When conservation gradients are calculated up to 30Å away, the average per-protein conservation-percolation correlation is higher, -0.6 (S2B Fig).

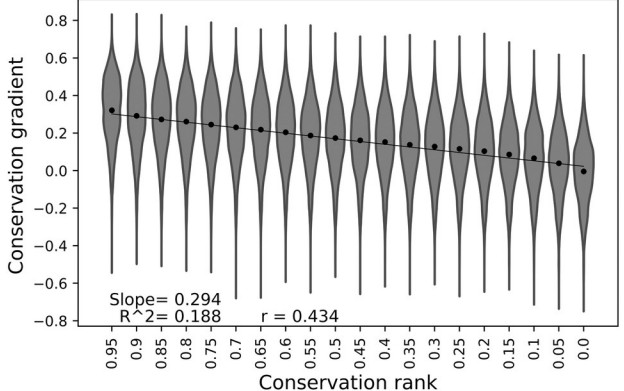
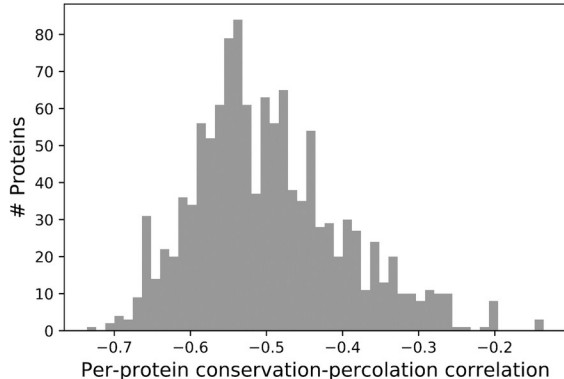

**Fig 1. Conservation gradient induced from a protein residue is linearly correlated with its conservation within the protein.** (A) Violin plots and respective average of conservation gradient from a residue as a function of conservation rank for all residues in the dataset binned into 20 equally spaced bins of conservation rank along with the linear fit calculated over all residues. (B) Distribution of per-protein Pearson correlation between residues' conservation ranks and conservation gradients.

We have also examined this relationship between site-specific conservation and site-induced conservation gradient specifically for functional sites. We identified different functional sites in the dataset: catalytic sites, non-catalytic ligand-binding sites (on enzymes and on nonenzymatic proteins), protein-protein interaction sites, and allosteric sites. We identified catalytic sites using the Mechanism and Catalytic Site Atlas (M-CSA)[20], ligand-binding sites using BioLip [21], allosteric sites using the Allosteric Database (ASD) [22–24] and protein-protein interaction sites using our previous protocol of identifying protein-protein interfaces in the yeast proteome[5]. Relative solvent accessibility (RSA) of residues was calculated and residues were classified as buried if RSA = 0.0, exposed if RSA>0.8 and middle for 0.0<RSA≤0.8. The linear relationship between site-specific conservation rank and site-induced conservation gradient holds regardless of the residue's location within the protein or its functional role (Fig 2 and S1 Table). Overall, our results reveal the existence of a conservation–percolation relationship in which higher residue conservation leads to stronger percolation of selective pressure to adjacent sites in the tertiary structure. Furthermore, this relationship holds for residues in different types of functional sites.

## Conservation gradient induced from a protein site is linearly correlated with the evolutionary rate of the site

The linear trend between conservation and percolation shown above is based on conservation ranks of residues, which are relative within a protein. In these calculations, we have lumped

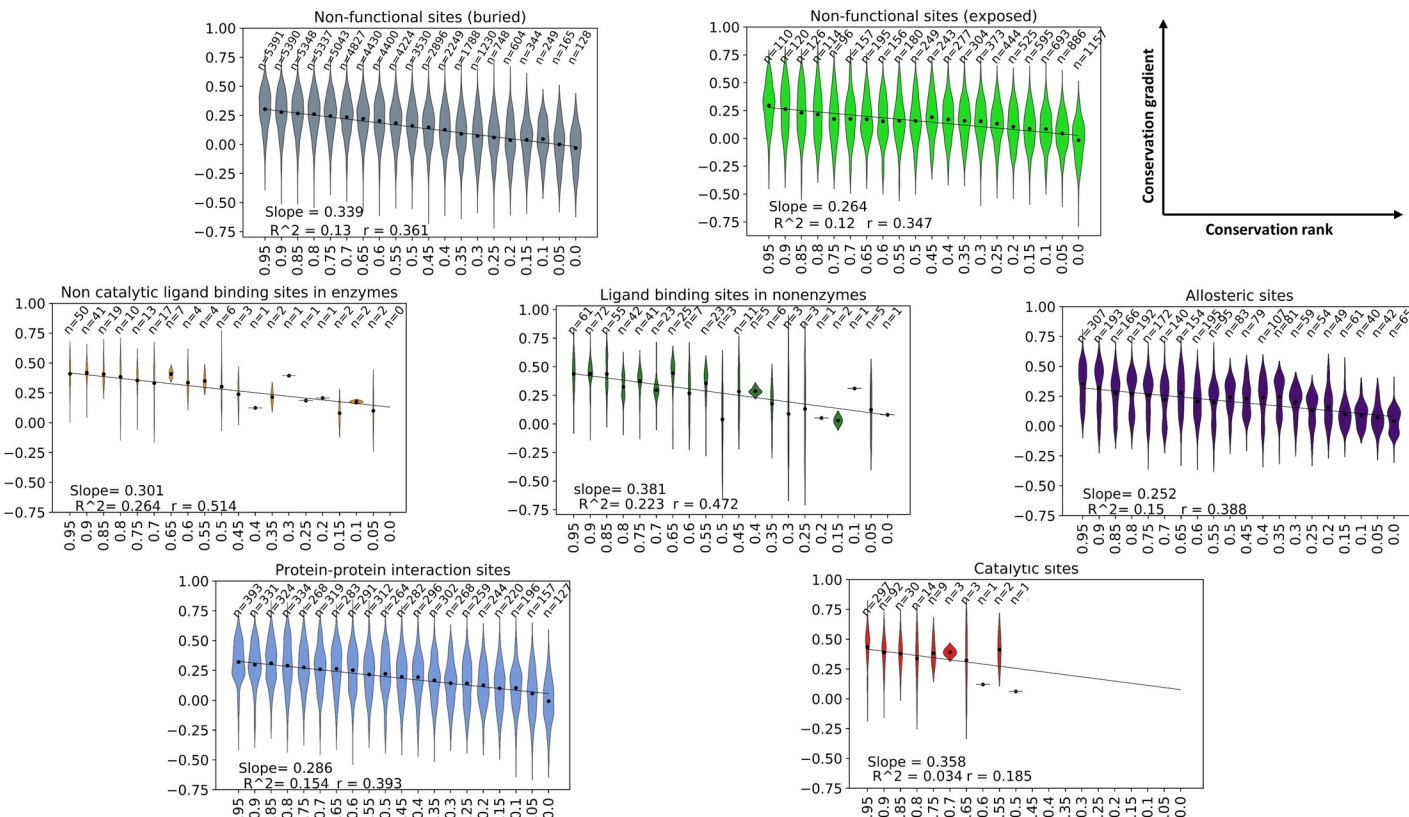

**Fig 2. Residues in functional and structural sites exhibit linear correlation between conservation and conservation gradient.** Violin plots of conservation gradient of residues as a function of their conservation rank within the protein along with the linear fit calculated over all residues in different types of structural and functional sites.

together residues from different proteins having similar conservation rank, however these residues, being from different proteins, can be under different selective pressure. To address this caveat, we examined whether the conservation-percolation linear trend still holds when conservation is measured in terms of absolute evolutionary rate (dN/dS). The evolutionary rate is calculated for the yeast proteins in *S. cerevisiae* compared with its orthologs in four closely related yeast species (see Methods). Conservation score annotations are transferred from structural homologs onto yeast proteins using sequence alignment. Then, we binned all residues into 100 equally spaced bins of conservation rank and calculated their average evolutionary rate (dN/dS). The average evolutionary rate increases for residues with increasing conservation rank (decreasing conservation) (S4 Fig), showing high correlation between the evolutionary rate of yeast protein sites and the conservation scores of their structural homologs. The conservation-percolation linear trend is shown to still hold here, where conservation is measured in terms of absolute evolutionary rate (Fig 3). The lower the average evolutionary rate of a protein site is, the stronger percolation of evolutionary conservation is induced from it.

Conservation gradients calculated as Spearman correlations exhibit a similar linear trend (S5 Fig) as well as conservation gradients calculated up to 30Å away from the reference residue (S6 and S7 Figs).

## Catalytic sites induce stronger conservation gradients than predicted by the conservation-percolation trend

The annotation of functional and structural site residues was transferred to yeast proteins from their structural models using sequence alignment. The average conservation gradient induced from them is shown in Fig 3A. As expected, functional residues evolve more slowly (low dN/dS) than other residues. The average conservation gradients induced from ligand binding sites in enzymes, allosteric sites and protein-protein interaction sites can all be predicted reasonably well from the linear conservation-percolation trend. Remarkably, catalytic

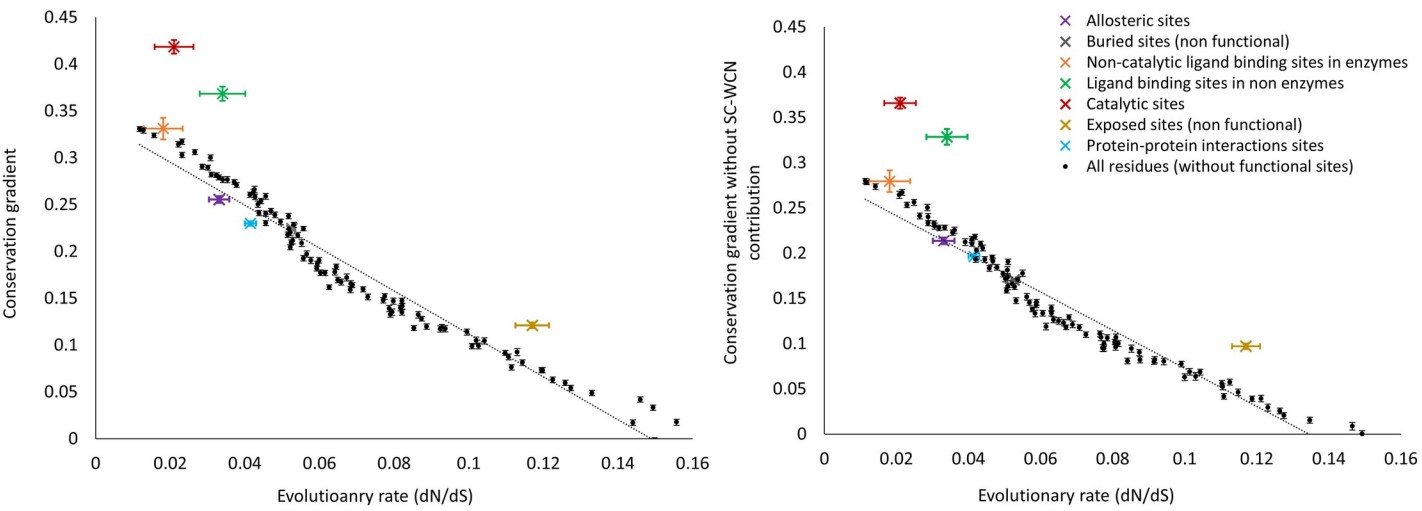

**Fig 3. Conservation gradient induced from residues is linearly (negatively) correlated with their evolutionary rate (dN/dS), with catalytic site residues inducing stronger conservation gradient than expected by the linear trend.** (A) Average conservation gradient (calculated as the average Pearson correlation between conservation of residues and their distance from a site) as a function of average evolutionary rate (dN/dS) for all yeast protein residues binned according to their annotated conservation rank into 100 equally spaced bins as well as the average conservation gradients of different types of functional sites. (B) Average conservation gradient without the relative contribution of SC-WCN as a function of average evolutionary rate (dN/dS) for all yeast protein residues binned according to their annotated conservation rank into 100 equally spaced bins as well as the average conservation gradients without the relative contribution of SC-WCN of different types of functional sites.

sites have the most significant deviation from the linear conservation-percolation trend. The average conservation gradient induced from catalytic site residues is significantly stronger than expected by the linear conservation-percolation trend. This can also be seen from the significantly stronger conservation gradients from catalytic sites compared with non-functional sites with similar high conservation rank (Fig 2). These results suggest that the strong conservation gradient induced from catalytic residues cannot be solely attributed to the percolation of the strong selective pressure on them. We repeated the analysis with the x-axis changed to the conservation rank (S8 Fig) showing agreement with Fig 3 supporting the conclusion that catalytic sites induce stronger conservation gradients on average than other sites with similar conservation. Ligand binding sites in nonenzymes in our dataset also induce somewhat stronger conservation gradients than expected (although not to the same extent as catalytic sites). This could be caused due to undiscovered catalytic sites and further work is required to test this hypothesis.

Conservation gradients calculated as Spearman correlations exhibit similar trends to those observed with conservation gradients expressed as Pearson correlations (S5 Fig). When conservation gradients measured up to 30Å (S6 and S7 Figs), the difference between those induced from catalytic sites and those induced from other sites with similar evolutionary rate is even higher compared with this difference when conservation gradients are calculated over the entire protein domain. Notably, conservation gradients up to 30Å from binding sites in nonenzymes are significantly lower compared with those from catalytic sites.

Comparing the conservation gradients from different types of functional sites, a possible caveat is that the residues composing them have a different variety of conservation ranks. While catalytic sites are smaller and contain mainly highly conserved residues, protein-protein interactions sites are larger and can include residues that have a small contribution to the function and are not highly conserved. We therefore repeated our analysis taking into account only the three most conserved residues from each functional site. These highly conserved residues have lower evolutionary rates than the entire functional site and induce higher conservation gradients (S9 Fig compared with Fig 3). The trend of the results is maintained showing that the most conserved residues within catalytic sites induce significantly stronger conservation gradients compared with the most conserved residues of other functional sites. Average conservation gradient from the most conserved residues of binding sites in nonenzymes is shown to be almost identical to that from catalytic sites.

The observed differences between conservation gradients induced from different functional sites could be dictated by structural determinants, such that tightly packed functional sites or those that are in a groove and hence close to the protein core, exhibit stronger conservation

**Table 1. Average side-chain weighted contact number (SC-WCN) and distance to the protein center as well as average conservation gradient of the different types of functional sites.**

| Functional sites | Mean SC-WCN | Mean distance to protein center | Mean conservation gradient |
|---|---|---|---|
| Catalytic sites | 2.62±0.03 | 7.1±0.4 | 0.41±0.01 |
| Non-catalytic ligand binding sites in enzymes | 2.65±0.03 | 9.3±0.4 | 0.30±0.01 |
| Ligand binding sites in nonenzymes | 1.92±0.02 | 7.4±0.3 | 0.37±0.01 |
| Protein-protein interaction sites | 1.70±0.01 | 14.8±0.1 | 0.21±0.00 |
| Allosteric sites | 2.03±0.01 | 10.8±0.2 | 0.24±0.00 |
| Buried non-functional sites# | 3.32±0.00[a] | 2.5±0.00[b] | 0.24±0.00[a]<br>0.20±0.00[b] |

# chosen as either

[a] sites for which SC-WCN>3.0

[b] sites for which distance to the protein center is <5.0

gradients. We have previously shown that any structural determinant of the protein backbone is unlikely to be a major determinant of the strong conservation gradients from enzymes[25]. We wanted to further test this hypothesis and control for site packing in our dataset. First, we have calculated the SC-WCN (side-chain weighted contact number)[9,10] for each residue in each protein in our dataset. SC-WCN is a measure of residue packing and centrality. While the average SC-WCN for catalytic sites is higher than for other functional sites, the average SC-WCN for catalytic sites is not significantly different from ligand binding sites in enzymes (Table 1). In addition, conservation gradients from non-functional buried sites with high SC-WCN induce significantly weaker conservation gradients than from other exposed functional sites such as protein-protein interaction sites. These results imply that packing does not dictate the difference in conservation gradients between these sites. Moreover, while the overall correlation coefficient between conservation gradients and conservation ranks over all the residues in our dataset is 0.43, the overall correlation coefficient between conservation gradients and SC-WCN values is significantly weaker (0.24).

We then constructed a linear regression model for conservation gradients of residues as a function of both their conservation rank and SC-WCN value. We subtracted the contribution of SC-WCN from the conservation gradient of every residue and plotted the new conservation gradients which are independent of SC-WCN (Fig 3B). The overall trends and differences in conservation gradients between different types of functional sites are maintained and are not strongly affected by controlling for the contribution of burial/packing. Similar results were obtained when the structural measure used was the proximity to the center of the protein (calculated as the distance from the residue with highest SC-WCN) (Table 1 and S10 Fig). Therefore, structural determinants of burial/packing or proximity of the functional site to the protein center are not the main cause of the significantly stronger conservation gradients from catalytic sites compared with non-catalytic sites.

## Catalytic site residues induce stronger conservation gradients than non-catalytic functional site residues with similar evolutionary rates

We have shown that on average, catalytic site residues induce stronger conservation gradients than expected by their average evolutionary rate. Next, we further reinforce this result by comparing the conservation gradient induced from subsets of catalytic and non-catalytic site residues with similar evolutionary rates. We have sampled 1,000 random subsets of residues from each type of functional site. For each such subset (represented by a circle in Fig 4A), we

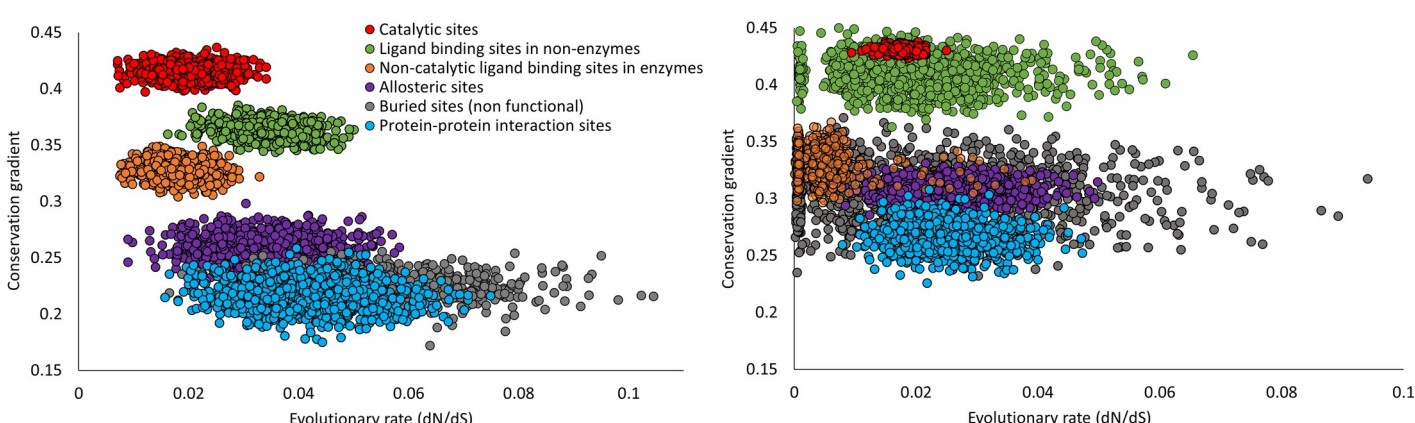

**Fig 4. Catalytic site residues induce stronger conservation gradients than non-catalytic functional site residues with similar evolutionary rates.** (A) Average conservation gradients of subsets of functional sites residues. (B) Average conservation gradients of subsets of the three most conserved residues within each functional site. Each circle represents a subset of residues, colored by different types of functional sites.

calculated the average evolutionary rate and average conservation gradient. As expected, catalytic site residue subsets span the lowest dN/dS values (x-axis), followed by non-catalytic ligand binding sites and allosteric sites, protein-protein interaction sites, ligand binding sites in nonenzymes and finally buried residues. The linear trend between evolutionary rate and conservation gradient holds for each of these functional site types (Fig 4A). This result indicates the robustness of the conservation-percolation linear trend regardless of the functional or structural role of the residues. Interestingly, residue subsets from different functional sites with similar evolutionary rates induce conservation gradients with different magnitudes. In particular, catalytic sites induce significantly stronger conservation gradients than all other non-catalytic sites with similar evolutionary rates, including non-catalytic ligand binding sites (Fig 4A). Notably, highly conserved buried nonfunctional residues that have similar evolutionary rates as those of catalytic sites, are shown here to induce significantly weaker conservation gradients (see also S11 Fig). This further shows that burial/packing of the functional site is not the main cause of the significantly stronger conservation gradients from catalytic sites compared with non-catalytic sites. Finally, protein-protein interaction site residues induce lower conservation gradients than most other functional site residues with similar evolutionary rates, possibly due to the tendency for protein-protein interactions to rewire during evolution.

We have repeated the analysis in Fig 4A with the x-axis changed to the conservation rank (S12 Fig). Results show broad agreement with Fig 4, supporting our main conclusion that catalytic sites induce stronger conservation gradient on average than other functional and nonfunctional sites, even after controlling for site-specific conservation level. In addition to catalytic sites, other ligand binding sites also exhibit somewhat higher conservation gradient than allosteric sites and protein-protein interaction sites, likely due to hidden, unannotated catalytic sites in our dataset of ligand binding sites.

When conservation gradients from subsets of residues are calculated with Spearman correlations, similar trends to those with Pearson correlations are obtained (S13 Fig). Moreover, conservation gradients up to 30Å exhibit similar trends to those calculated over the entire protein domain (S14 and S15 Figs). Even though conservation gradients up to 30Å are generally smaller in magnitude, the large difference between those induced from catalytic sites and those from non-catalytic functional sites with similar evolutionary rate is even more pronounced than conservation gradients computed over the protein domain.

When considering only the three most conserved residues from each functional site (Fig 4B), subsets of residues exhibit lower evolutionary rates and higher conservation gradients compared with subsets from all functional sites residues (Fig 4A). Interestingly, even subsets of non-catalytic sites residues with same or lower evolutionary rates than catalytic sites residues induce significantly lower conservation gradients. These results further emphasize the unique behavior of catalytic sites that induce significantly stronger conservation gradients than other sites with similar evolutionary conservation and cannot be completely explained by their low evolutionary rates. Conservation gradients from ligand binding sites in nonenzymes are mostly lower than those from catalytic sites although some induce similar conservation gradients which might be caused due to possible 'hidden catalytic sites'.

## Catalytic site residues often induce stronger conservation gradients than more conserved non-catalytic functional site residues within the same protein

We have shown that conservation gradients induced by functional site residues in general correlate linearly with the evolutionary rates of these functional site residues. We have also shown that catalytic site residues are special in that they induce the strongest conservation gradients

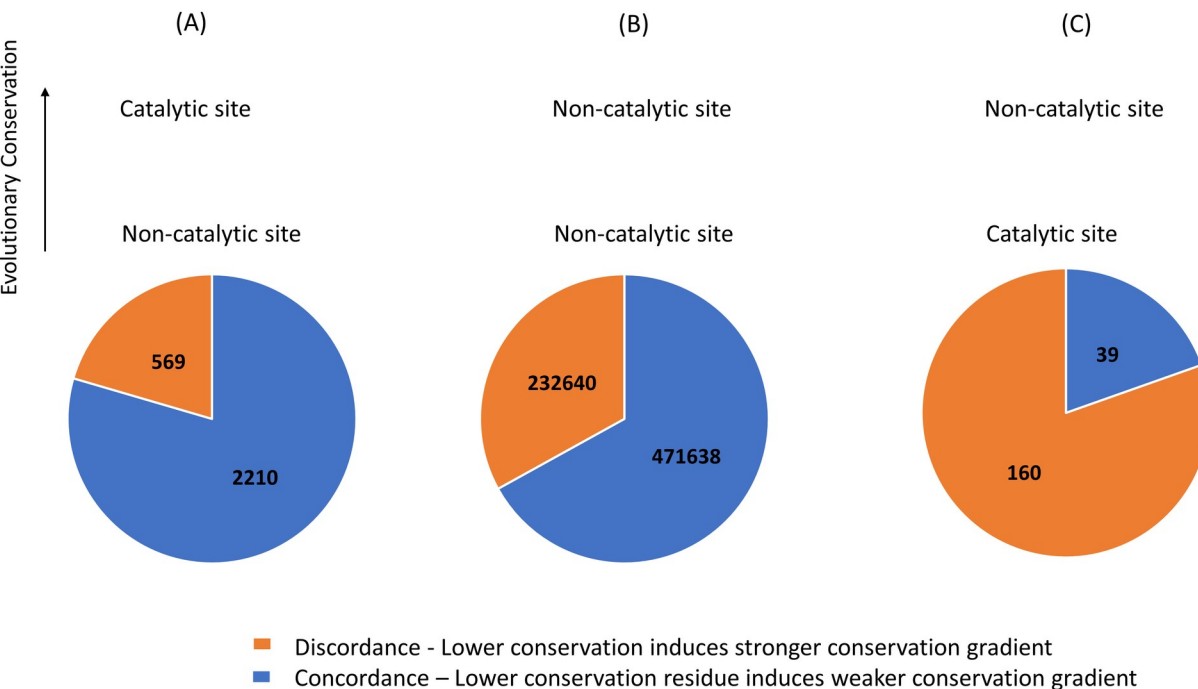

**Fig 5. Catalytic site residues often induce stronger conservation gradients than more conserved non-catalytic functional site residues within the same protein.** Within the same protein, (A) more conserved catalytic site residues tend to induce stronger conservation gradient than less conserved non-catalytic site residues (binomial test, P<<0.001); (B) more conserved non-catalytic site residues tend to induce stronger conservation gradient than less conserved non-catalytic site residues (binomial test, P <<0.001); (C) less conserved catalytic site residues often induce stronger conservation gradient than more conserved non-catalytic site residues (binomial test, P<<0.001). Functional site residue pairs for which the ordering of residue conservation agrees with the ordering of induced conservation gradient (concordance) are marked in blue. Functional site residue pairs for which the ordering of residue conservation disagrees with the ordering of induced conservation gradient (discordance) are marked in orange.

among subsets of different functional site residues with similar average evolutionary rates. However, our analyses have so far been carried out by grouping together functional site residues from different proteins. In this section, we perform a stringent, per-protein analysis by focusing on multi-functional proteins with at least two distinct functional sites and comparing the evolutionary properties of different functional site residues within the same protein. Some proteins in our dataset contain both catalytic sites as well as non-catalytic functional sites, whereas other proteins in our dataset contain two distinct non-catalytic functional sites. We found that for the majority of the functional site residue pairs within the same multi-functional protein, the more conserved functional site residue indeed induces stronger conservation gradient (80% if the more conserved residue is catalytic, Fig 5A; 67% if both residues are non-catalytic, Fig 5B). These results agree with the hypothesis that induced conservation gradients are largely driven by the percolation of selective pressure acting on functional sites.

Remarkably, in cases where the catalytic site residue is less conserved than the non-catalytic functional site residue within the same multi-functional protein, the catalytic site residue still induces stronger conservation gradient than the non-catalytic functional site for most of these cases (discordance of 80%, Fig 5C). This large discordance shown in cases where the lower conservation residue is a catalytic site residue, is significantly higher compared to the cases where the lower conservation residue is a non-catalytic site residue (binomial test P<<0.01). Similar trends are obtained with conservation gradients calculated as Spearman correlations (S16 Fig) as well as when the analysis is focused only on the three most conserved residues from each functional site (S17 Fig) and when conservation gradients are computed up to 30Å (S18 Fig).

These results clearly show that within the same protein, less conserved catalytic site residues often induce stronger conservation gradient than the more conserved non-catalytic site. Therefore, the strong conservation gradients from catalytic sites cannot be entirely explained by the percolation of the strong selective pressure acting on the catalytic sites.

## Discussion

In this paper we have shown a linear relationship between two measures of evolutionary importance of protein sites under purifying selection. These are the degree of evolutionary conservation of the site itself, as well as the percolation of evolutionary conservation induced from the site via neighboring residues in the protein tertiary structure. Despite major differences between these two measures, we have shown that the linear relationship between the two measures is universal as it holds for different types of proteins as well as for different types of functional sites in proteins. However, catalytic sites in enzymes are the principal exception to this rule. We have shown here that catalytic sites in enzymes induce significantly stronger conservation gradients in the rest of the protein than expected from the degree of conservation of the site alone. Catalytic sites have a unique and complex functionality as they both bind a substrate as well as reduce the free energy barrier required for a chemical reaction to occur. These catalytic sites were shown to be under stronger selective pressure compared with other functional sites such as protein-protein interaction sites and ligand binding sites[13,15]. It was also shown that they induce a significantly stronger evolutionary rate gradient than other functional sites. One hypothesis regarding the origin of the strong conservation gradient from catalytic sites is that it is simply due to the strong selective pressure acting on these sites percolating through the rest of the protein via residue-residue contacts. However, we have shown here that the strong selective pressure acting on catalytic sites cannot entirely explain the strong conservation gradients induced from catalytic sites.

The main determinant of the conservation gradient from catalytic sites is still not completely understood. Local structural constraints (such as residue burial and packing, WCN) are usually potential contributors as they are known to generally have a significant effect on residue evolutionary rate[1,2,5,10]. However, it was shown that generally, local structural constraints are not the main determinants of conservation gradients in enzymes[13,25,26]. Moreover, the fact that non-catalytic ligand binding sites and allosteric binding sites induce significantly weaker evolutionary rate gradients implies that the ligand binding and allosteric function are not the main determinants of the conservation gradients in enzymes either[15]. We are therefore left with the hypothesis that the uniquely strong conservation gradient in enzymes is imposed by the special requirement for catalytic sites to differentially bind the transition state of a chemical reaction rather than the reactants or products with very similar properties [16], a function which is unique to catalytic sites compared with other functional binding sites. A recent physical model of residue evolutionary rates in enzymes introduced an activation term in addition to a stability term and showed improved predictive ability[27]. The model attributed the activity to the free energy required to transform a distorted catalytic site upon mutation to its native conformation. The improved ability of the model supports the hypothesis that the main determinant of the observed conservation gradient in enzymes is a functional rather than structural constraint. Overall, our results suggest that the stringent requirement for the catalytic site to differentially bind to and stabilize the transition state of the catalyzed chemical reaction imposes extensive evolutionary constraints on a large portion of the enzyme beyond just the catalytic site, all of which play key roles in maintaining the catalytic function.

Accurate quantification of selective pressure on proteins at single-site resolution is an important task in evolutionary biology [2]. We have shown here that there are two different

methods to quantify the evolutionary importance of a protein residue. The classical, "intrinsic" measure of conservation and the "extrinsic" measure which is conservation gradient the site exerts on the rest of the protein. The combination of these two measures provides a complete, quantitative picture of evolutionary conservation patterns within proteins induced by functional sites. The linear relationship between the degree of conservation of a protein functional site and the induced conservation gradient in the rest of the protein suggests that the strong selective pressure acting on the functional site percolates through the rest of the protein via residue-residue contacts. Our results also clearly show the unique evolutionary behavior of enzymes in which the catalytic site induces significantly stronger evolutionary constraints on their surroundings than can be explained by the percolation theory alone. Moreover, our results emphasize that catalysis requires the participation of a much larger set of residues than just the few key catalytic residues.

The current study is empirical, using available data on annotated functional sites and their conservation gradient patterns. In future work it will be interesting to use simulation lattice models[14] or biophysical models[27] to examine the effect of different factors on conservation gradient patterns and to unify the empirical and theoretical studies.

## Methods

### Protein dataset collection and functional site annotations

The current dataset is based on a dataset of structural homologs of yeast proteins[15]. The dataset was created first by using gapped BLAST[28] searches between protein subunit sequences with solved structure from the Protein Data Bank[17] and 5,861 translated open reading frames (ORFs) of the yeast *Saccharomyces cerevisiae*[29]. The ORF–subunit pairs were chosen such that both the subunit sequence and the ORF sequence had coverage of $\geq$50% in the alignment and E-value $<10^{-5}$ and could be paired with their orthologs in four other closely-related yeast species *S. paradoxus*, *S. mikatae*, *S. bayanus* and *S. pombe*. This way, 1,555 yeast ORFs were mapped to homologs in the PDB. The procedure included the following steps:

First, if one of the homology-based structural models of a yeast ORF had an annotated allosteric site, this model was chosen. For 171 yeast proteins the structural model was identified with a known allosteric site as well as pre-calculated conservation scores in ConSurf-DB[18]. For all other yeast ORFs, if they had structural models with known ligand binding sites that do not overlap with catalytic sites, the model with the lowest E-value out of them was chosen. Overall, 39 nonenzymes with 42 ligand binding sites and 20 enzymes with 25 non-catalytic ligand-binding sites were part of the dataset. For all other yeast ORFs, the structural model with the lowest E-value was chosen. In this manner, 976 more ORFs for which the best structural model had pre-calculated conservation scores in ConSurf-DB were added to the dataset. Overall, 1,206 ORF-subunit pairs were included in the study. Out of them, 147 yeast proteins were identified with 282 protein-protein interactions sites and 107 proteins were identified with catalytic sites. Full list of yeast proteins, their structural models along with identified functional sites can be found in S2 Table in the Supporting Material.

### Functional site annotations

Allosteric sites within the structural subunits were found and their residues annotated using the Allosteric Database (ASD)[22–24]. Biologically-relevant ligand-binding sites were found using Binding MOAD (Mother of all Databases)[30,31], a database of biologically significant protein-ligand binding in the PDB. Using MOAD, sites with bound crystallographic additives, buffers, salts, metals and sites with covalently linked ligands are excluded. Ligand-binding

residues were identified using the BioLip[21] database. Catalytic sites within the protein structural subunits were found using the M-CSA [20], taking into account also all protein chains in the PDB which are more than 95% identical to protein chains found in M-CSA[32]. In order to find proteins that participate in protein-protein interactions in our dataset, we identified structural subunits where each subunit is both in physical contact with another subunit and the corresponding modelled ORFs are reported as interacting by at least one physical experiment in the BioGRID [33,34]. Our dataset contains 147 proteins with 282 protein-protein interaction interfaces. Interfacial residues were identified as residues with different solvent accessibility values when in complex compared to when the interacting partner is manually deleted from the tertiary structure. Distances between residues were calculated as distances between their respective Cα atoms. All functional site annotations of the chosen structural subunits were transferred to the yeast ORF sequence according to the sequence alignment.

## Evolutionary conservation and rate calculations

In this study we calculated both evolutionary conservation scores for the residues of the structural subunits as well as average absolute evolutionary rates (dN/dS) for the yeast ORFs. Evolutionary conservation scores were taken from ConSurf-DB[18], which is a database of pre-calculated conservation scores of residues in proteins with known structures in the Protein Data Bank (PDB). ConSurf-DB conservation scores are based on collected sequence homologs of the PDB structure and using the Rate4Site algorithm[35]. S1 Text in the Supporting Material provides all conservation scores obtained from ConSurf-DB for all the proteins used in this study. Calculated conservation gradients for each residue in every protein in the dataset can all be found in Supporting S2 Text. S2 Text also lists the conservation gradients calculated using Spearman correlation, calculated up to 30Å away from the reference residue and calculated with the relative contribution of SC-WCN eliminated.

To calculate the average evolutionary rates (dN/dS) for residues of *S. cerevisiae*, we first used the orthology assignment of the protein-coding genes of *S. cerevisiae* with four other closely-related yeast species (*S. paradoxus, S. mikatae, S. bayanus, and S. pombe)*, according to the Fungal Orthogroup Repository[36]. We then aligned the ORFs using MAFFT[37]. Then, evolutionary rates were calculated using the program codeml within the PAML software package[38]. The tree was specified as ((((*S. cerevisiae, S. paradoxus), S. mikatae), S. bayanus), S. pombe*). Codon frequencies were assumed equal (CodonFreq = 0) and other parameters in codeml were left to their default values. The codon alignments can be found in S3 Text in the Supporting Material.

## Statistical analysis

1000 random sample of 250 residues were collected from each type of functional site residues for Fig 4.

Estimated standard errors in our measurements of conservation gradients (Pearson correlations) and of dN/dS values were done using 50 rounds of bootstrap resampling.

## Supporting information

**S1 Fig. Conservation gradient induced from a protein residue is linearly correlated with its conservation within the protein.** (A) Violin plots and respective average of conservation gradient (calculated as a Spearman correlation between a residue conservation and its distance from a site) as a function of conservation rank for all residues in the dataset binned into 20 equally spaced bins of conservation rank along with the linear fit calculated over all residues. (B) Distribution of per-protein Pearson correlation between residues' conservation ranks and

conservation gradients (conservation gradients calculated as Spearman correlations).
(TIF)

**S2 Fig. Conservation gradient induced from a protein residue is linearly correlated with its conservation within the protein.** (A) Violin plots and respective average of conservation gradient (calculated as a Pearson correlation between a residue conservation and its distance from a site up to 30Å away) as a function of conservation rank for all residues in the dataset binned into 20 equally spaced bins of conservation rank along with the linear fit calculated over all residues. (B) Distribution of per-protein Pearson correlation between residues' conservation ranks and conservation gradients (conservation gradients calculated as Pearson correlations up to 30Å away).
(TIF)

**S3 Fig. Conservation gradient induced from a protein residue is linearly correlated with its conservation within the protein.** (A) Violin plots and respective average of conservation gradient (calculated as a Pearson correlation between a residue conservation and its distance from a site between 6 Å and 30Å away) as a function of conservation rank for all residues in the dataset binned into 20 equally spaced bins of conservation rank along with the linear fit calculated over all residues. (B) Distribution of per-protein Pearson correlation between residues' conservation ranks and conservation gradients (conservation gradients calculated as Pearson correlations for between 6 Å and 30Å away).
(TIF)

**S4 Fig. Evolutionary rate is correlated with conservation rank.** Evolutionary rate (dN/dS) as a function of conservation rank for all residues in the dataset grouped according to their conservation rank and binned into 100 equally spaced bins of conservation rank.
(TIF)

**S5 Fig. Conservation gradient induced from residues is linearly (negatively) correlated with their evolutionary rate (dN/dS), with catalytic site residues inducing stronger conservation gradient than expected by the linear trend.** Average conservation gradient (calculated as the average Spearman correlation between conservation of residues and their distance from a site) as a function of the average evolutionary rate (dN/dS) for all yeast protein residues binned according to their annotated conservation rank into 100 equally spaced bins (black) as well as the average conservation gradients of different types of functional sites.
(TIF)

**S6 Fig. Conservation gradient induced from residues is linearly (negatively) correlated with their evolutionary rate (dN/dS), with catalytic site residues inducing stronger conservation gradient than expected by the linear trend.** Average conservation gradient (calculated as the average Pearson correlation between conservation of residues and their distance from a site up to 30Å away) as a function of the average evolutionary rate (dN/dS) for all yeast protein residues binned according to their annotated conservation rank into 100 equally spaced bins (black) as well as the average conservation gradients of different types of functional sites.
(TIF)

**S7 Fig. Conservation gradient induced from residues is linearly (negatively) correlated with their evolutionary rate (dN/dS), with catalytic site residues inducing stronger conservation gradient than expected by the linear trend.** Average conservation gradient (calculated as the average Pearson correlation between conservation of residues and their distance from a site between 6Å and 30Å away) as a function of the average evolutionary rate (dN/dS) for all yeast protein residues binned according to their annotated conservation rank into 100 equally

spaced bins (black) as well as the average conservation gradients of different types of functional sites.
(TIF)

**S8 Fig. Conservation gradient induced from residues is linearly (negatively) correlated with their evolutionary conservation, with catalytic site residues inducing stronger conservation gradient than expected by the linear trend.** Average conservation gradient as a function of average conservation rank for all yeast protein residues binned into 100 equally spaced bins as well as the average conservation gradients of different types of functional sites.
(TIF)

**S9 Fig. Conservation gradient induced from residues is linearly (negatively) correlated with their evolutionary rate (dN/dS), with catalytic site residues inducing stronger conservation gradient than expected by the linear trend.** Average conservation gradients (calculated as the average Pearson correlation between conservation of residues and their distance from a site) as a function of the average evolutionary rate (dN/dS) for all yeast protein residues binned according to their annotated conservation rank into 100 equally spaced bins (black) as well as the average conservation gradients of the three most-conserved residues within each functional site.
(TIF)

**S10 Fig. Average conservation gradient after reduction of the relative contribution of proximity to the protein center, as a function of average evolutionary rate (dN/dS) for all yeast protein residues binned according to their annotated conservation rank into 100 equally spaced bins as well as the average conservation gradients without the relative contribution of proximity to the protein center of different types of functional sites.**
(TIF)

**S11 Fig. Average conservation gradients of subsets of buried and exposed, non-functional site residues.** Each circle represents a subset of residues.
(TIF)

**S12 Fig. Average conservation gradients of functional sites binned according to conservation rank up to 0.65.**
(TIF)

**S13 Fig. Catalytic site residues induce stronger conservation gradients than non-catalytic functional site residues with similar evolutionary rates.** Average conservation gradients (calculated as the average Spearman correlation between conservation of residues and their distance from a site). Each circle represents a subset of residues, coloured by the different types of functional sites.
(TIF)

**S14 Fig. Catalytic site residues induce stronger conservation gradients than non-catalytic functional site residues with similar evolutionary rates.** Average conservation gradients (calculated as the average Pearson correlation between conservation of residues and their distance from a site up to 30Å away). Each circle represents a subset of residues, coloured by the different types of functional sites.
(TIF)

**S15 Fig. Catalytic site residues induce stronger conservation gradients than non-catalytic functional site residues with similar evolutionary rates.** Average conservation gradients (calculated as the average Pearson correlation between conservation of residues and their distance

from a site between 6Å and 30Å away). Each circle represents a subset of residues, coloured by the different types of functional sites.
(TIF)

**S16 Fig. Catalytic site residues often induce stronger conservation gradients than more conserved non-catalytic functional site residues within the same protein.** Within the same protein, when conservation gradients are calculated as Spearman correlation between conservation of residues and their distance from a site (A) more conserved catalytic site residues tend to induce stronger conservation gradient than less conserved non-catalytic site residues (binomial test, P<<0.001); (B) more conserved non-catalytic site residues tend to induce stronger conservation gradient than less conserved non-catalytic site residues (binomial test, P <<0.001); (C) less conserved catalytic site residues often induce stronger conservation gradient than more conserved non-catalytic site residues (binomial test, P<<0.001). Functional site residue pairs for which the ordering of residue conservation agrees with the ordering of induced conservation gradient (concordance) are marked in blue. Functional site residue pairs for which the ordering of residue conservation disagrees with the ordering of induced conservation gradient (discordance) are marked in orange.
(TIF)

**S17 Fig. Catalytic site residues often induce stronger conservation gradients than more conserved non-catalytic functional site residues within the same protein.** Within the same protein, considering only conservation gradients from the three most-conserved residues within each functional site (A) more conserved catalytic site residues tend to induce stronger conservation gradient than less conserved non-catalytic site residues (binomial test, P<<0.001); (B) more conserved non-catalytic site residues tend to induce stronger conservation gradient than less conserved non-catalytic site residues (binomial test, P <<0.001); (C) less conserved catalytic site residues often induce stronger conservation gradient than more conserved non-catalytic site residues (binomial test, P<0.001). Functional site residue pairs for which the ordering of residue conservation agrees with the ordering of induced conservation gradient (concordance) are marked in blue. Functional site residue pairs for which the ordering of residue conservation disagrees with the ordering of induced conservation gradient (discordance) are marked in orange.
(TIF)

**S18 Fig. Catalytic site residues often induce stronger conservation gradients than more conserved non-catalytic functional site residues within the same protein.** Within the same protein, when conservation gradients are calculated as Pearson correlation between conservation of residues and their distance from a site up to 30Å away (A) more conserved catalytic site residues tend to induce stronger conservation gradient than less conserved non-catalytic site residues (binomial test, P<<0.001); (B) more conserved non-catalytic site residues tend to induce stronger conservation gradient than less conserved non-catalytic site residues (binomial test, P <<0.001); (C) less conserved catalytic site residues often induce stronger conservation gradient than more conserved non-catalytic site residues (binomial test, P<<0.001). Functional site residue pairs for which the ordering of residue conservation agrees with the ordering of induced conservation gradient (concordance) are marked in blue. Functional site residue pairs for which the ordering of residue conservation disagrees with the ordering of induced conservation gradient (discordance) are marked in orange.
(TIF)

**S1 Table. Linear regression of conservation gradients as a function of conservation rank over all residues in different types of functional and structural categories.**
(XLSX)

**S2 Table. List of all yeast proteins, their structural models and identified functional sites.**
(XLSX)

**S1 Text. Conservation scores downloaded from ConSurf-DB for all the proteins participating in this study.**
(RAR)

**S2 Text. Calculated conservation gradients for each residue in every protein in the dataset.**
The file also lists the conservation gradients calculated using Spearman correlation, calculated up to 30Å away from the reference residue and calculated with the relative contribution of SC-WCN eliminated.
(RAR)

**S3 Text. Codon alignments of protein coding genes of *S. cerevisiae* with its four closely-related yeast species (*S. paradoxus*, *S. mikatae*, *S. bayanus*, and *S. pombe*).**
(RAR)

## Author Contributions

**Conceptualization:** Avital Sharir-Ivry, Yu Xia.

**Formal analysis:** Avital Sharir-Ivry.

**Supervision:** Yu Xia.

**Writing – original draft:** Avital Sharir-Ivry.

**Writing – review & editing:** Avital Sharir-Ivry, Yu Xia.

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
