## [Decision Letter · Decision Letter 0]

7 Sep 2020

Dear Dr Sharir-Ivry,

Thank you very much for submitting your Research Article entitled 'Quantifying Evolutionary Importance of Protein Sites: A Tale of Two Measures' to PLOS Genetics. Your manuscript was fully evaluated at the editorial level and by two independent peer reviewers. The reviewers appreciated the attention to an important problem, but raised some substantial concerns about the current manuscript. Based on the reviews, we will not be able to accept this version of the manuscript, but we would be willing to review again a much-revised version. We cannot, of course, promise publication at that time.

If you decide to revise the manuscript for further consideration at PLOS Genetics, please aim to resubmit within the next 60 days, unless it will take extra time to address the concerns of the reviewers, in which case we would appreciate an expected resubmission date by email to plosgenetics@plos.org.

[LINK]

We are sorry that we cannot be more positive about your manuscript at this stage. Please do not hesitate to contact us if you have any concerns or questions.

Yours sincerely,

Jianzhi Zhang

Associate Editor

PLOS Genetics

Kirsten Bomblies

Section Editor: Evolution

PLOS Genetics

Reviewer's Responses to Questions

**Comments to the Authors:**

Reviewer #1: In the manuscript “Quantifying Evolutionary Importance of Protein Sites: A Tale of Two Measures”, the authors reported the relationship between the evolutionary constraint of a site in protein and its conservation gradient. For a site, its gradient is the pearson correlation between the constraints on the other sites and their distances to the focal site. A stronger gradient of a site (larger correlation) indicates that its neighboring sites tend to have stronger constraints than the distal sites. The authors observed a linear relationship, suggesting that the sites with strong constraints also had strong conservation gradients. The authors concluded that such a relationship was likely due to residue-residue contacts among the sites etc. Particularly, the authors found that catalytic sites in enzymes had much stronger conservation gradients than the other sites, and these strong gradients could not be explained by the particularly strong constraints on the focal sites. Therefore, the authors concluded that the observation was likely caused by the catalytic function. Although the results and conclusions are interesting, I have several major concerns.

Major points

One main result of the manuscript is that the sites with strong constraints tend to have strong conservation gradients. This linear relationship is convincing and expected. It is known that neighboring residues are expected to be involved in the same biological function e.g. catalysis, binding, and protein stability etc., and thus tend to have similar constraints. For example, in the reference 13, the residues in close proximity to the strongly constrained residues also have strong constraints, whereas the distal residues are less constrained. Therefore, the strongly constrained residues are expected to have relatively larger gradients than the residues with weak constraints, resulting in the observed relationship. However, different from what the author claimed, I think the linear relationship is likely quite weak, given the large variance of each bin. And it worth reporting more details on the regression, e.g. R-square and whether only the median/mean of each bin was used for the regression, which artificially reduces the data variation.

Another major result is that the catalytic residues have particularly strong conservation gradients. The authors concluded that the high conservation gradients are probably because the unique requirement for the active site to selectively stabilize the transition state of the catalyzed chemical reaction imposes additional selective constraints on the rest of the enzyme. However, the large gradients of catalytic sites may be because all the identified catalytic sites in a catalytic region have strong constraints, whereas taking the PPI sites as an example, it is known that the PPI interface is relatively large but only a few key PPI residues have strong constraints, and the neighboring sites have weak constraints. This renders even the key residues having low conservation gradients. In addition, catalytic sites tend to be buried rather than exposed on protein surfaces as other binding sites. Located at the core region of a protein further increases conservation gradients due to the paths from core (high constraints) to surface (very low constraints). In sum, without controlling for these factors, the high gradients of the catalytic sites may not be due to the intrinsic catalytic properties.

It may be interesting to quantify the influences of all these factors using a simple lattice model. The functional sites, core sites and surface sites have their constraints sampled from respective constraint distributions to calculate gradients. The factors may include the size of the functional region in the protein which has core and surface regions, the average and variance of site constraints in the functional region, the location of the region (surface or core/grove) etc.

Overall, the gradients are moderate or small, calculated using Pearson correlation. Spearman correlation robust to outliers may be necessary to confirm the discovery.

Minor points

The authors mentioned very briefly that their discovery is important to phylogenetic inference, accurate quantification of selective pressure at single-site resolution etc. Please discuss a bit more the details in the discussion.

The authors used “long-range” conservation in the manuscript. It would be useful to define the long range e.g. up to 30A. However, there is a possibility that many of the general conservation gradients observed by the authors are mainly due to “short” range residues.

In this manuscript, many sites from different proteins were pooled together to estimate an average dn/ds for these sites. Many of those sites may have quite different dn/ds. PAML may be used to test whether the sites in a protein have different evolutionary rates, and then estimate the rates respectively for the sites. The multiple groups of sites with different rates may be informative for the analyses.

Reviewer #2: Overall, this is a nice contribution. However, I have one major concern: Most proteins have a natural conservation gradient from the outside to the inside. So any study trying to identify some alternative cause for a conservation gradient must very carefully control for this strong confounder. I don't think the present study does so. I would argue the present study doesn't even properly discuss this issue.

To me, the key question is to what extent sites create a conservation gradient given where they are in the protein structure. The authors look at buried and exposed sites, but that's a very crude classification. A site can be buried but relatively close to the surface or right in the center of the protein, and these two sites will experience both different selection pressures and different conservation gradients.

A good measure to assess how close a site is to the center of the protein core is the weighted contact number (WCN), using an inverse square distance weighting. In fact, WCN is literally a measure of centrality, rather than a measure of number of contacts. (As an aside, many authors in the field mis-understand this issue.) If the authors correlate WCN with conservation gradient, they should find a fairly strong correlation. Then, the authors can build a regression model that regresses the conservation gradient against both WCN and conservation rank. The degree to which conservation rank contributes to such a model is a measure of the intrinsic conservation rank a site generates, independent of where in the structure it is located. It may well be that if the authors perform this analysis, catalytic sites stand out even more.

**Have all data underlying the figures and results presented in the manuscript been provided?**

Reviewer #1: Yes

Reviewer #2: **No: **I think the authors should provide their raw data and analysis scripts. As is, the study is not reproducible.

PLOS authors have the option to publish the peer review history of their article (what does this mean?). If published, this will include your full peer review and any attached files.

Reviewer #1: No

Reviewer #2: No

---

## [Decision Letter · Decision Letter 1]

6 Jan 2021

Dear Dr Xia,

Thank you very much for submitting your Research Article entitled 'Quantifying Evolutionary Importance of Protein Sites: A Tale of Two Measures' to PLOS Genetics.

The manuscript was fully evaluated at the editorial level and by the two original peer reviewers. While one reviewer is fully satisfied, the other has a few comments that need to be addressed.

We therefore ask you to modify the manuscript according to the review recommendations. Your revisions should address the specific points made by each reviewer.

[LINK]

Yours sincerely,

Jianzhi Zhang

Associate Editor

PLOS Genetics

Kirsten Bomblies

Section Editor: Evolution

PLOS Genetics

Reviewer's Responses to Questions

**Comments to the Authors:**

Reviewer #1: In this revision, the authors added more results and analyses, and the conclusions are more solid. However, there are still several concerns.

In the manuscript, the authors emphasize “long-range” gradients. I commented on this before suggesting the authors to define long vs short. In this revision, the authors analyzed the residues within 30A. I feel that to support “long-range”, the author should have analyzed the residues beyond a certain cutoff (within a shell) for catalytic residues and other residues.

I suggested that the strong correlation between constraint and conservation gradient may be due to the location of catalytic residues, i.e. close to the core of proteins. (The core-surface and catalytic function together lead to the high gradient). To address this comment, the authors used the numbers of contact residues to indicate the packing. However, catalytic residues (other ligand binding residues) may be in a groove (thus close to core) but have no contact residues. Number of contact residues may not be a direct measure for this purpose.

Fig2 shows the relationship between the normalized conservation ranks of residues and their conservation gradients, for different types of residues, such as ligand binding sites, catalytic sites, allosteric sites etc. The conclusion is that the relationship for catalytic sites is quite unique. However, from the fig, it seems that this is likely because the catalytic sites have x range 0.95 to 0.65. For some other residue types, the relationships in this range seem similar to that of catalytic sites. The authors may need to add regression lines using only that x range. The r-square (cor squared) is quite small, indicating the fitting is at most moderate.

For fig4, are the black dots “all residues (w/o functional sites)”? Their results are missing in panel B. The result of such residues can tell how much the constraints on residues alone influence the gradients.

Conservation gradients depend on the relative residue constraints within each protein. The normalization used by fig1&2 is more reasonable than comparing dn/ds from different proteins. It seems that the normalized conservations can be used for those key analyses in fig3&4 with x changed accordingly.

About the conclusion in DIscussion, I think the measures in this manuscript probably can not be informative for “de-novo functional site prediction and protein design”, because many functional sites, except catalytic sites, are similar to non-functional sites in terms of the measures.

The following sentences may contain typos.

When considering only the three most conserved residues from each functional site (Fig 4B), subset of residues exhibits lower evolutionary rates and higher conservation gradients compared with subsets from all functional sites residues (Fig 4A).

Beyond the classical, “intrinsic” measure of conservation and the “extrinsic” measure which is conservation gradient the site exerts on the rest of the protein.

Reviewer #2: Thank you for your careful revisions. I have no further comments.

**Have all data underlying the figures and results presented in the manuscript been provided?**

Reviewer #1: Yes

Reviewer #2: Yes

PLOS authors have the option to publish the peer review history of their article (what does this mean?). If published, this will include your full peer review and any attached files.

Reviewer #1: No

Reviewer #2: No

---

## [Decision Letter · Decision Letter 2]

9 Mar 2021

Dear Dr Xia,

We are pleased to inform you that your manuscript entitled "Quantifying Evolutionary Importance of Protein Sites: A Tale of Two Measures" has been editorially accepted for publication in PLOS Genetics. Congratulations!

Yours sincerely,

Jianzhi Zhang

Associate Editor

PLOS Genetics

Kirsten Bomblies

Section Editor: Evolution

PLOS Genetics

Comments from the reviewers (if applicable):

Reviewer's Responses to Questions

**Comments to the Authors:**

Reviewer #1: The authors addressed my comments

**Have all data underlying the figures and results presented in the manuscript been provided?**

Reviewer #1: Yes

PLOS authors have the option to publish the peer review history of their article (what does this mean?). If published, this will include your full peer review and any attached files.

Reviewer #1: No

**Data Deposition**

http://datadryad.org/submit?journalID=pgenetics&manu=PGENETICS-D-20-01275R2

**Press Queries**

---

## [Editor Report · Acceptance letter]

23 Mar 2021

PGENETICS-D-20-01275R2 

Quantifying Evolutionary Importance of Protein Sites: A Tale of Two Measures 

Dear Dr Xia, 

We are pleased to inform you that your manuscript entitled "Quantifying Evolutionary Importance of Protein Sites: A Tale of Two Measures" has been formally accepted for publication in PLOS Genetics! Your manuscript is now with our production department and you will be notified of the publication date in due course.

With kind regards,

Alice Ellingham

PLOS Genetics

On behalf of:
